# OpenReview forum: "SciVideoBench: Benchmarking Scientific Video Reasoning in Large Multimodal Models"
_ICML.cc/2026/Conference — ICML 2026 regular_

### Official Review · Reviewer_17jR · 2026-02-23

**Soundness:** 3
**Presentation:** 3
**Significance:** 3
**Originality:** 3
**Overall Recommendation:** 5
**Confidence:** 4

**Summary:**

This paper introduces SciVideoBench, a benchmark specifically designed to assess advanced video reasoning in scientific contexts. Different from existing benchmarks that focus on college-level science videos, SciVideoBench focuses on research-level science videos. The authors evaluate various LLMs and humans on their benchmark and analyze the existing challenges.

**Compliance With Llm Reviewing Policy:**

Affirmed.

**Final Justification:**

The authors addressed my concerns. I think it's a novel work for evaluating the scientific reasoning ability of LLMs.

**Key Questions For Authors:**

1. Why are only these four disciplines adopted in the benchmark? How about other disciplines, such as Mathematics, Geography, Materials, and Electronics?
2. It's unclear whether the knowledge required for the question is all contained in the video.
3. The authors haven't mentioned whether the audio is utilized in the evaluation.
4. Generally, AI4S methods use search engine APIs to obtain external knowledge. Do you use search engine APIs or the built-in search tools of proprietary models?

**Limitations:**

yes

**Strengths And Weaknesses:**

Strengths:
1. The proposed benchmark extends science video reasoning from college-leval to research-level, providing a timely evaluation protocol for AI4S.
2. The questions in the dataset are hard, verified by the low accuracy by graduate students.
3. The source of videos is reliable and the bias  in annotation is well addressed.

Weaknesses:
1. While there are various science disciplines, the authors only adopt four disciplines. The evaluation is incomplete.
2. It's unclear whether the knowledge required for the question is all contained in the video. Ideally, this should be ensured in the benchmark.
3. The authors haven't mentioned whether the audio is utilized in the evaluation. If not, the evaluation may be biased.
4. Generally, AI4S methods use search engine APIs to obtain external knowledge. I suggest including the experiments with search engine APIs.

---

> ### Author Rebuttal · Authors · 2026-03-31
>
> ## 1. Only four disciplines
>
> We thank the reviewer for this question. SciVideoBench is intentionally scoped to experimental, laboratory-based videos, where scientific reasoning requires interpreting observable procedures, measurements, and outcomes from visual evidence. As a result, disciplines such as mathematics or geography, where problems are typically abstract, symbolic, or not grounded in laboratory experiments, are not included in the current benchmark.
>
> At the same time, the four selected domains (Biology, Chemistry, Physics, and Medicine) already cover a broad range of experimental sciences. As shown in Figure 3, SciVideoBench includes over 25 subfields (e.g., materials science, fluid dynamics, nanomaterials, bioengineering), capturing substantial diversity in experimental setups and reasoning types.
>
> Therefore, our goal is not to exhaustively cover all disciplines, but to focus on expert-level procedural and methodological reasoning in experimental settings, which these domains naturally support.
>
>
> ## 2. Whether required knowledge is contained in the video
>
> We ensure that the key information required to answer each question is grounded in the video through multiple stages. First, the Visual Comparer verifies that all necessary visual cues (e.g., measurements, experimental outcomes, or observable changes) are present in the video. Subsequently, human annotators further validate that each question is answerable based on the video content.
>
> This multi-stage process ensures that the benchmark evaluates visual-grounded reasoning rather than reliance on external or implicit knowledge.
>
>
> ## 3. Audio utilization
>
> We thank the reviewer for this question. In the main evaluation, we focus on visual input and do not include audio, as SciVideoBench is designed to emphasize visual-grounded scientific reasoning, where models must first identify critical visual cues from the video before performing downstream reasoning.
>
> To assess the role of audio, we additionally provide modality ablation results in Appendix N and Appendix K. These results consistently show that while audio-only signals can provide complementary information (e.g., narrated explanations), performance remains significantly lower than video-based reasoning.
>
> In Appendix K, enabling audio leads to only modest improvements for models that support full multimodal input (e.g., +2.7% for Gemini-2.5-Pro and +2.8% for Qwen2.5-Omni-7B), further indicating that audio provides limited additional gains. Moreover, combining audio with video yields consistent but incremental improvements, suggesting that audio acts as a complementary modality rather than a primary signal.
>
> Overall, these findings demonstrate that model performance is predominantly driven by visual content, and that our evaluation—while excluding audio in the main setting—does not introduce bias but instead isolates the role of visual-grounded reasoning.
>
> Therefore, excluding audio in the main evaluation does not introduce bias, but instead isolates the role of visual understanding, which is the primary focus of SciVideoBench.
>
>
> ## 4. Use of search engine APIs
>
> We thank the reviewer for this suggestion. In this work, we adopt a closed-book evaluation setting and focus on assessing the model’s intrinsic visual perception and reasoning ability, rather than its ability to retrieve external knowledge via search APIs or tools.
>
> We note that retrieval in this context would primarily provide access to the associated research papers. To evaluate its potential impact, we include a **video+paper** setting in our modality ablation (Table 11). The results show only a modest improvement over video-only QA (24.9 → 25.8), indicating that access to external textual information provides limited additional benefit.
>
> This is consistent with our task design: answering SciVideoBench questions requires extracting **instance-specific visual evidence** from the video, which cannot be recovered from external documents alone. Therefore, retrieval does not fundamentally address the core challenge of the benchmark.
>
> We acknowledge that in some benchmarks (e.g., Humanity’s Last Exam), enabling search tools leads to noticeable improvements, as these tasks often involve retrieving relevant knowledge or facts. In contrast, SciVideoBench emphasizes **visual-grounded reasoning over experimental processes**, where the key bottleneck lies in perception rather than knowledge access.

---

> > ### Author Rebuttal · Reviewer_17jR · 2026-04-02
> >
> > Thanks for the authors' response. I have the following concerns:
> > 1. According to your response, it's more appropriate to claim that this benchmark is for reasoning on laboratory experiments, not for general scientific reasoning. Therefore, the presentation (e.g., the title) may be problematic.
> > 2. While the authors show in the appendix that adding audio only brings little accuracy improvements, there is no ablation study of only using the audio input. Therefore, I don't think the current results can support the conclusion that model performance is predominantly driven by visual content.
> >
> > While some improvements to this work should be done, I still recognize the value of this benchmark that requires very zero-shot scientific reasoning ability.

---

> > > ### Author Response · Authors · 2026-04-05
> > >
> > > We thank the reviewer for the thoughtful follow-up and for recognizing the value of our benchmark.
> > >
> > > We agree that SciVideoBench focuses on reasoning over laboratory experiments. However, we would like to clarify that this is a deliberate and appropriate definition of scientific video reasoning. **In the video modality, scientific reasoning is fundamentally grounded in experimental processes**, where knowledge is demonstrated, observed, and validated through real-world procedures and outcomes.
> > >
> > > In contrast, disciplines such as mathematics or theoretical physics primarily involve symbolic or abstract reasoning, which are not naturally expressed or evaluated through videos. As a result, they fall outside the scope of scientific **video** reasoning. Therefore, focusing on laboratory-based experimental scenarios is not a restriction, but rather a necessary condition for defining meaningful scientific reasoning in video form.
> > >
> > > Under this definition, we believe the current scope and naming of SciVideoBench appropriately reflect the nature of the task. We will further clarify this formulation in the paper to avoid potential misunderstanding.
> > >
> > > Regarding the role of audio, we would like to clarify that we **do include audio-only ablation in Appendix N** (Table 12). Results with Gemini-2.5-Pro show that audio-only performance (34.9%) is substantially lower than video-only performance (64.3%), demonstrating that audio input alone is insufficient for solving the benchmark even for strong models. Furthermore, while audio provides complementary contextual information, combining audio with video yields only moderate improvements (67.0%), indicating that visual evidence remains the dominant signal for accurate reasoning.
> > >
> > > Additionally, we provide detailed audio ablation results for VITA-1.5 and Qwen-2.5-Omni to further support this conclusion:
> > >
> > > | Model         | Setting      | Overall | Conceptual | Hypothetical | Quantitative |
> > > |--------------|-------------|---------|------------|--------------|--------------|
> > > | VITA-1.5     | Audio-only  | 11.7    | 13.2       | 14.8         | 4.6          |
> > > | VITA-1.5     | Audio+Video | **20.7**    | **28.1**       | 21.3         | 8.6          |
> > > | VITA-1.5     | Video-only  | 18.4    | 21.6       | **22.1**         | 7.8          |
> > > | Qwen-2.5-Omni| Audio-only  | 9.3     | 10.0       | 11.0         | 7.5          |
> > > | Qwen-2.5-Omni| Audio+Video | **17.5**    | **20.3**       | **18.5**         | **12.8**         |
> > > | Qwen-2.5-Omni| Video-only  | 14.7    | 16.7       | 15.0         | 9.7          |
> > >
> > > These results consistently show that audio provides complementary information but is insufficient on its own, further supporting that visual grounding is the primary driver of performance. We will clarify this more explicitly in the main text to avoid confusion.

---

### Official Review · Reviewer_Nsnv · 2026-03-03

**Soundness:** 3
**Presentation:** 3
**Significance:** 2
**Originality:** 2
**Overall Recommendation:** 3
**Confidence:** 3

**Summary:**

This paper introduces SciVideoBench, a new benchmark that was designed to evaluate LMM on research-level science video reasoning and understanding. SciVideoBench contains 1,000 multiple-choice questions that was derived from 241 videos of experimental dataset JoVE, covers physics, chemistry, biology, and medicine domains including more than 60 different sub-domains.

Questions are split into three types. Conceptual questions are evaluate fact related reasoning on steps, hypothetical questions focus on reasoning with assumptions, and quantitative questions use calculations through functions to receive the result.

The author(s) ensure the question quality through an human-AI collaborative annotation process that combines human experts (graduate students) with multiple LLM agents (QA Generator, Evaluator, Visual Comparer, and Refiner) and two AI question filters that will remove questions that do not require video input to figure out the result.

The paper evaluates over 38 close and open-source LMMs of various sizes, finding that even the best-performing model (Gemini-2.5-Pro) only achieves 64.3% accuracy, compared to a human expert baseline of just 17.4%. This highlights the significant gap between current models and expert-level scientific reasoning.

The paper also provides an extensive analysis of model performance across metrics including reasoning type, subject domain, model scaling, chain-of-thought prompting, audio input, and frame sampling, with detailed failure case analyses (as shown in Figures 7, 14, 15, 16).

**Compliance With Llm Reviewing Policy:**

Affirmed.

**Final Justification:**

My concerns are partly resolved, and I decide remain my score. I hope the authors can provide more substantial experiment results to strengthen this paper.

**Key Questions For Authors:**

1. Please use specific examples to elaborate on the fundamental differences between questions in SciVideoBench and those in recent scientific video benchmarks like MMVU and Video-MMMU.
2. In the annotation pipeline, how many questions were filtered out by the "Visual Comparer" and "Visual Blind Test"? Could you provide results from a human verification of a random sample (e.g., 50 questions) from the final 1000 to demonstrate that these questions indeed cannot be answered using transcripts or the accompanying manuscript text alone?
3. Regarding the explanation for open-source models' performance decline on conceptual/hypothetical reasoning with CoT, could you provide a deeper analysis?
4. Were the participating graduate students experts in the specific experiments shown in the videos? Was their number of video viewings limited? Could the score (17.4%), which is much lower than intuitive expectations, be are lower than expectation due to the small sample size?
5. The paper states that CoT reduces the performance of open-source models on conceptual and hypothetical reasoning. Please provide an deeper analysis of the specific reasons for this phenomenon.
6. Please provide a fine-grained error analysis of why multimodal models fail on "Quantitative" tasks? Is it because models fail to accurately locate physical parameters in the video, or because they lack scientific calculation capabilities?
7. Given that the current dataset relies entirely on standardized JoVE videos, might models be taking shortcuts by learning JoVE-specific narrative and editing patterns?

**Limitations:**

The author(s) explicitly discuss limitations in Appendix S (Limitation), covering aspects like data source coverage (primarily JoVE, English-only materials), potential annotation bias, and the benchmark's primary focus on the lab setting. Appendix R (Ethical Considerations) provides a detailed discussion of ethical considerations, data privacy, dual-use risks, etc. Overall, the authors provide a candid and thorough discussion of limitations.

**Strengths And Weaknesses:**

Strengths：

1. The benchmark construction process is good, featuring well-designed multi-stage human-AI collaborative annotation, visual blind testing, and human verification, which are well designed and effectively ensuring data quality.
2. The paper has a clear overall structure, logical flow and coherent progress. Figures and tables effectively aid in understanding the core ideas. The appendix provides rich and detailed supplementary information.

Weakness:

1. Although this annotating framework uses AI to filter and double check whether questions require video to answer, there is no human secondary verification or cross-validation through multiple AIs.
2. The paper needs to differentiate SciVideoBench more clearly from recent benchmarks also targeting scientific/educational videos (e.g., MMVU, Video-MMMU). This should be done by directly comparing example questions to demonstrate its unique aspects in reasoning depth and visual dependency, rather than simply labeling older benchmarks as "college-level" and its own as "research-level." The data source JoVE primarily showcases established experimental methods. Describing it as "research-level" might lack precision. A more accurate description might be "expert-level procedural and methodological reasoning."
3. Originality is tempered by the recent emergence of similar works, making the direction less purely novel. The heavy reliance on LLMs (especially Gemini 2.5 Pro) in the annotation process, despite human verification, necessitates a more careful discussion and quantification of their role as assistants and potential bias risks within the methodology design.

Other Concerns:
1. Reference formatting is inconsistent, with heavy reliance on arXiv preprints and inconsistent journal/conference abbreviations.
2. The annotation flowchart (Figure 2) lacks rigor, with inconsistencies between the figure content and the main text (including the following errors):

- Implied inconsistency: The figure suggests that human Annotated Example QAs are part of the prompt input to the AI, rather than human Annotated Example QAs expanded by other AIs being part of the prompt input.
- Lack of clear modular introduction: The phrase "In the second stage" makes it difficult to quickly locate the relevant part in the figure.
- Unlabeled symbols: When only viewing the figure, cannot directly identify the symbol to the left of "evaluator" as "manuscript text" or the symbol to its right as "video." It also fails to show the characteristic that both symbols represent inputs to the Evaluator.
- Text-figure inconsistency: The relationship between "Review and Add Critical Details" and "Refiner" is described more as sequential in the section but appears parallel in the figure.
- Missing components in the figure: The two types of "GPT-4o double check sections" are not shown in the diagram. The experimental results table (Table 2) suffers from poor readability. Although it simplified from Table 5, it still can be split into multiple separate tables for clearer comparison (e.g., open-source vs. proprietary models, with/without CoT, comparisons within the same open-source model series) to present a more intuitive result. The table does not explicitly state that the metric is accuracy, which could easily lead to confusion.

---

> ### Author Rebuttal · Authors · 2026-03-31
>
> ## 1. Comparison with related benchmarks
>
> MMVU includes questions such as “What could the brown stuff in the video be?” and “Assume that 2.24 liters ...; how many grams of precipitate are produced?”, which emphasize entity recognition or reasoning under pre-assumptions, etc. Video-MMMU mainly uses lecture-style videos, where key information is often conveyed through textual or OCR cues rather than complex real-world scenes. By contrast, SciVideoBench is constructed from real experimental recordings, where critical information must be inferred from the ovservation in the video.
>
> Both are explicitly described as “college-level” in their papers (MMVU Table 1; Video-MMMU Sec. 3.1). This difference is also reflected in human performance(MMVU 49.7%, Video-MMMU 74.44% (senior undergraduates), SciVideoBench 17.4% (graduate students)). Moreover, in Table 7, SciVideoBench exhibits long reasoning depth in both step count and reasoning length. Finally, JoVE videos are derived from peer-reviewed experimental research, involving complex setups and domain-specific procedures rather than simplified instructional content. We will include explicit example-based comparisons in the revision.
>
>
>
> ## 2. Filtered questions
>
> About 7,000 questions are initially generated; after Visual Comparer, ~4,000 (Appendix F) are left. GPT-4o further reduces to ~1,000, followed by final human verification.
>
> We conduct a human study (4 volunteers) on 36 questions from 8 papers. Participants mark questions as “unsolvable” if the manuscript text is insufficient. We find that 35/36 are unsolvable using paper-only information, with the only exception being a quantitative case where a key parameter appears in a figure reflecting the video setup.
>
> This confirms that SciVideoBench cannot be reliably solved from textual input alone. Consistent with our design, conceptual and hypothetical questions require inspecting specific timestamps (Figure 4), while quantitative questions depend on extracting key parameters directly from the video.
>
>
> ## 3. Open-source CoT
>
> Quantitative questions benefit more from CoT, as they require multi-step computation. Conceptual and hypothetical questions rely more heavily on accurate visual grounding, scientific knowledge, and causal reasoning. For open-source MLLMs, CoT may amplify errors by encouraging elaboration on incorrect visual perception or unstable knowledge, leading to error accumulation (Fig.14, InternVL-3-14B) and “overthinking.” By contrast, proprietary models are better able to maintain visual grounding throughout the reasoning process.
>
> This observation aligns with prior work (Mind Your Step (by Step): Chain-of-Thought can Reduce Performance on Tasks where Thinking Makes Humans Worse, ICML 2025), which shows that CoT can degrade performance when intermediate reasoning steps are unreliable, particularly in perception-heavy tasks.
>
>
> ## 4. Graduate students in human evaluation
>
> The graduate students were not involved in the videos and therefore did not have prior familiarity with the exact setups. To ensure broad coverage, we recruited participants based on disciplinary background and assigned each a fixed subset of videos. Each question was evaluated once per participant in a single-pass setting.
>
>
> ## 5. Quantitative failure analysis
>
> To analyze failure modes in quantitative reasoning, we conduct a detailed study using Gemini-2.0-Flash. We compare CoT outputs with human-verified rationale and timestamps, and identify three main error types: visual recognition errors (70.7%), where the model fails to correctly extract key visual evidence; calculation errors (17.2%), where quantities are identified but incorrectly combined; and background knowledge errors (8.1%), where incorrect scientific understanding leads to faulty reasoning, with the remaining 4% corresponding to other issues such as correct reasoning but incorrect final answers. We will include further analysis for other question types in the revision.
>
>
> ## 6. JoVE shortcuts
>
> SciVideoBench is constructed from recently published JoVE videos, reducing potential overlap with model pretraining data. The dataset also spans over 25 subfields, covering diverse experimental setups and environments. More importantly, our questions require extracting instance-specific visual evidence that cannot be inferred from narrative or editing patterns alone.
>
>
> ## 7. Other concerns
>
> We will address presentation issues in the revision as follows: (1) standardize reference formatting; (2) clarify that GPT-4o-expanded annotated examples are used and update Figure 2 accordingly; (3) improve figure modular clarity and alignment with the text; (4) add labels for all symbols (e.g., manuscript text, video inputs); (5) revise the workflow to consistently reflect the intended pipeline (Refiner → human verification); (6) incorporate the GPT-4o double-check stages into the diagram; and (7) improve Table 2 readability by explicitly stating the metric (accuracy) and reorganizing results.

---

> > ### Author Rebuttal · Reviewer_Nsnv · 2026-04-02
> >
> > Thank you for the detailed rebuttal. My concerns are partly resolved, and I decide remain my score. I hope the authors can provide more substantial experiment results to strengthen this paper.

---

> > > ### Author Response · Authors · 2026-04-03
> > >
> > > We thank the reviewer for the feedback and for acknowledging that our rebuttal has partially addressed the concerns.
> > >
> > > We would like to clarify that we have addressed the main points raised in the review with additional analysis and evidence. Specifically:
> > >
> > > - **Q1 & W2 (benchmark comparison)**: We provide explicit comparisons with MMVU and Video-MMMU, including example questions, differences in video content, human performance, and reasoning depth.
> > > - **Q2 (text-only solvability)**: We conduct an additional human study on a random subset, showing that the questions cannot be reliably solved using textual information alone.
> > > - **Q3 & Q5 (CoT behavior)**: We analyze CoT and show that performance degradation in open-source models is mainly due to weak visual perception, where CoT amplifies perception errors (e.g., misinterpreting spectral peaks in Fig. 14).
> > > - **Q4 (human evaluation setup)**: We clarify that the human evaluators are not involved in the experimental videos.
> > > - **Q6 (quantitative failure analysis)**: We provide a detailed analysis using Gemini-2.0-Flash, showing that the dominant failure source is visual perception rather than calculation.
> > > - **Q7 (JoVE shortcuts)**: We explain that using recent JoVE videos and diverse experimental domains mitigates potential shortcut learning.
> > > - **Other concerns**: We will address all remaining presentation issues in the revision.
> > >
> > >
> > > To better understand how we can further strengthen the paper, we would greatly appreciate if the reviewer could kindly clarify which aspects of the experimental evaluation you feel remain insufficient. In particular, it would be helpful to know whether additional experiments on specific settings (e.g., CoT analysis, modality ablation, or failure analysis) would help address the remaining concerns.
> > >
> > > Thanks!

---

### Official Review · Reviewer_rrsS · 2026-03-07

**Soundness:** 2
**Presentation:** 2
**Significance:** 2
**Originality:** 3
**Overall Recommendation:** 3
**Confidence:** 3

**Summary:**

This paper introduces SciVideoBench, a benchmark for scientific video reasoning.
It contains 1,000 multiple-choice questions derived from 241 research-level experimental videos. Questions are categorized into conceptual, hypothetical, and quantitative reasoning. The authors evaluated 38 Large Multimodal Models (LMMs). Results show that even top models like Gemini 2.5 Pro and GPT-4o struggle significantly.

**Compliance With Llm Reviewing Policy:**

Affirmed.

**Final Justification:**

The authors still not justify how necessary the video is and the potential contamination as future models internalize the knowledge in the video. In general it's the limitation of all the static benchmarks.

**Key Questions For Authors:**

see cons

**Limitations:**

see cons

**Strengths And Weaknesses:**

# Pros:
1. We indeed dont have video QA benchmarks yet; it's good to have one.
2. The questions seem challenging. Even grad students only hit 17.4%, which is a refreshing sanity check.
3. Semi-automatic pipeline with multi-stage human verification is reasonably rigorous
4. Comprehensive model sweep (38 models) with useful ablations (audio, frame sampling, CoT)


# Cons:
1. To be honest I feel the video per se is not that needed for the QA? And as time goes by, models with the latest knowledge cut-off (say they have been trained on the papers can easily answer the questions without watching the video).

Also, the modality ablation (Table 11) shows transcript-only GPT-4o gets 21.5% vs video-only at 24.9%. I know their performances are low alrady but still there's only a tiny gap. Audio-only Gemini hits 34.9%. The "video is essential" claim is quite weak empirically.

Fig 1 probably shows the same thing. If the rest data in the benchmark all look like that, a smart reader (or LLM) with the transcript could often reconstruct the answer without watching anything? Here we need strong justification.

2. Sourcing exclusively from JoVE creates domain/language bias and potential paywalling reproducibility issues.

---

> ### Author Rebuttal · Authors · 2026-03-31
>
> ## 1. Video Necessity
>
> This concern is not supported by our empirical results. The relatively small gap between transcript-only and video-only performance for GPT-4o reflects the model’s limited ability to fully exploit visual information, rather than the irrelevance of video.
>
> To clarify the role of video in our benchmark, SciVideoBench is explicitly designed to require **visual perception as a prerequisite for reasoning**, with different roles across question types:
>
> - **Conceptual and Hypothetical reasoning** usually require models to interpret specific experimental steps and outcomes at precise moments in the video. As illustrated in Figure 4, questions explicitly provide timestamps, forcing the model to inspect visual details (e.g., procedures, state changes, or interactions) before reasoning about mechanisms or counterfactual scenarios.
>
> - **Quantitative reasoning** requires models to first locate and extract key parameters from the video (e.g., measurements, quantities, or experimental conditions), as shown in Figure 1, and then perform multi-step reasoning or calculation based on these visually grounded values.
>
> Given this design, video perception is not optional but a necessary step for solving the questions.
>
> Empirically, this is further supported by results from Gemini-2.5-Pro, where video-only performance (64.3%) significantly exceeds audio-only performance (34.9%) by a large margin (+29.4). Moreover, combining audio and video further improves performance, indicating that modalities provide complementary signals.
>
> These findings suggest that while textual information offers useful context, accurate performance fundamentally depends on grounded visual understanding when models are capable of utilizing it. This interpretation is also supported by Reviewer 4oDm, who notes that our modality ablations demonstrate that the dataset largely resists text-only solution paths while remaining driven by visual evidence.
>
>
> ## 2. Transcript vs. On-screen text
>
> We thank the reviewer for this observation and would like to clarify a potential misunderstanding. The text shown in Figure 1 is not part of the transcript, but rather on-screen text embedded in the video, which serves as a form of visual evidence (e.g., numerical values or experimental measurements). Such information is only accessible through visual perception rather than textual input.
>
> In the original JoVE videos, on-screen text is often used when certain measurements or quantities are not clearly observable from the raw footage. In cases where such critical visual cues are missing, we explicitly annotate them as on-screen overlays to ensure that key information remains grounded in the visual modality.
>
> Therefore, answering these questions still requires visual understanding, rather than relying on transcript-based reasoning.
>
> ## 3. JoVE source: domain bias and reproducibility
>
> This concern is not supported by the characteristics of our dataset. We argue that using JoVE does not inherently introduce domain or language bias, and we clarify the reproducibility considerations as follows:
>
> - **Diverse institutional and geographic sources**: JoVE videos are produced by independent research groups across universities and institutes worldwide (241 videos/papers with 881 unique affiliations), rather than a single curated pipeline. This reduces the risk of domain-specific or stylistic bias.
>
> - **Broad scientific and experimental coverage**: SciVideoBench spans four major disciplines with multiple specialized subfields (e.g., bioengineering, materials science, analytical chemistry, oncology, fluid dynamics), each involving distinct experimental paradigms, observation targets, and reasoning requirements.
>
> - **Heterogeneity in experimental and visual distributions**: The videos exhibit substantial variation in instrumentation, procedures, laboratory environments, narration styles, and camera setups, reflecting real-world diversity rather than a fixed distribution.
>
> - **Language bias is limited**: Although JoVE videos are primarily in English, the benchmark focuses on **visual-grounded reasoning over experimental processes**, where key information (e.g., measurements, procedural steps, visual outcomes) is conveyed through visual evidence rather than linguistic variation.
>
> - **Reproducibility considerations**: While JoVE is a curated source, our benchmark construction process (including question annotations, timestamps, and evaluation protocols) is fully specified and reproducible. We will further clarify data access and usage details in the final version to ensure transparency and facilitate future research.
>
> Overall, JoVE provides a high-quality and diverse collection of *peer-reviewed experimental videos*, enabling evaluation of visual-grounded reasoning across heterogeneous real-world scientific scenarios, rather than introducing systematic bias.

---

> > ### Author Rebuttal · Reviewer_rrsS · 2026-04-01
> >
> > Thanks for the clarification. I will adjust my score to weak reject as the authors still not justify how necessary the video is and the potential contamination as future models internalize the knowledge in the video. In general it's the limitation of all the static benchmarks.

---

> > > ### Author Response · Authors · 2026-04-03
> > >
> > > We thank the reviewer for the follow-up and for reconsidering the score.
> > >
> > > Regarding the necessity of video, we emphasize that SciVideoBench is explicitly designed to require visual grounding across different question types, as described in our response. This is further supported by our modality ablation results. In addition, the Qwen model blind-test results provide complementary evidence: VL models outperform their non-VL counterparts by approximately **+7.4%** on average, demonstrating the importance of video input.
> > >
> > > Regarding the concern of potential contamination as models internalize knowledge over time, we agree that this is a general limitation shared by all static benchmarks. To mitigate this, we construct SciVideoBench using recently published JoVE videos, **reducing overlap with the pretraining data of evaluated models**. In particular, most videos are published after March 2025, while the majority of evaluated models (Table 4) were released before February 2025, making direct memorization unlikely. Furthermore, JoVE videos are not freely crawlable like open platforms (e.g., YouTube) due to copyright restrictions, which further **limits large-scale inclusion in pretraining corpora**.
> > >
> > > We agree that long-term robustness requires continuous updates. As JoVE is an actively updated source, we plan to maintain SciVideoBench as **a living benchmark by incorporating newly published videos**, thereby reducing potential contamination and preserving evaluation validity over time.

---

### Official Review · Reviewer_4oDm · 2026-03-12

**Soundness:** 3
**Presentation:** 4
**Significance:** 4
**Originality:** 3
**Overall Recommendation:** 5
**Confidence:** 3

**Summary:**

The ‌submission ‌presents ‌SCIVIDEOBENCH, a set of 1,000 multiple-choice items derived from 241 research-grade experimental videos taken from JoVE. Coverage spans physics, chemistry, biology, and medicine. Answering the questions typically requires tracking visual information over time, bringing relevant scientific knowledge to bear, and carrying out multi-step reasoning rather than relying on a single cue. Across evaluations of more than 38 large multimodal models under several test conditions, including direct answering, chain-of-thought prompting, vision-blind variants, and audio removal, the main finding is that performance remains limited. The top result is reported for Gemini-2.5-Pro at 64.3% overall accuracy, and the quantitative category appears to be the most difficult. For data construction, the authors rely on a multi-agent LLM workflow paired with substantial human verification, and they include a GPT-4o-based vision-blind check intended to remove questions that are not genuinely grounded in the video.

**Compliance With Llm Reviewing Policy:**

Affirmed.

**Key Questions For Authors:**

Distractors for quantitative questions are generated by adding Gaussian noise to the correct answer (Appendix C). Given that quantitative scores are already near random for many models (e.g., InternVL-3-1B-Instruct at 7.76%), is it possible that some distractors are physically implausible (e.g., negative masses, impossibly large volumes), allowing partial elimination? An analysis of whether the 9.39% floor in Appendix I reflects genuine difficulty or distractor artifacts would strengthen these results.

**Limitations:**

The authors adequately discuss limitations in Appendix S, including JoVE sourcing bias, potential annotation errors, and scope restricted to laboratory scenarios. The ethical considerations and copyright treatment are also carefully handled. However, the limitation regarding the human baseline validity concern noted above is not addressed, and the potential circularity of using GPT-4o as both a data filter and an evaluated model deserves explicit acknowledgment. The authors should add these points to Section S.

**Strengths And Weaknesses:**

A clear strength is the three-part taxonomy of question types, conceptual, hypothetical, and quantitative, which is argued coherently and aligns with distinct cognitive requirements. Empirically, the types separate in useful ways, including different chain-of-thought benefits as reported in Section 4.3. The construction process also appears careful, with the vision-blind filtering step in Section 3.2 and explicit attention to choice-length balance in Figure 13, both of which speak to common shortcuts and annotation artifacts in earlier benchmarks. The modality ablations in Appendix N are another solid component, because they disentangle the roles of video, transcript, audio, and the associated paper, and they suggest that the dataset largely resists text-only solution paths while remaining driven by visual evidence. The benchmark also seems to discriminate effectively among systems, leaving noticeable headroom even for the strongest model, and the reported chain-of-thought gains on SCIVIDEOBENCH, for example, +14% for Gemini-2.0-Flash, are larger than those cited for MMVU (+3%) and VideoMathQA (+6.6%), which supports the claim that the task is demanding.

At the same time, several issues limit interpretability. All videos are drawn from a single source, JoVE, which plausibly introduces stylistic regularities and a constrained domain distribution. Although Section S acknowledges the risk, the paper does not examine how within-source imbalance might influence the headline metric. Biology comprises 38.9% of the videos, while Medicine is 12.8% (Figure 3), yet it remains unclear whether the aggregate accuracy is disproportionately shaped by the majority category. A second concern comes from the human baseline: Table 2 reports 17.4% accuracy for graduate students in a closed-book setting, only modestly above 10% random choice. While positioned as evidence of difficulty, this result also raises a validity question, namely, whether a nontrivial portion of the benchmark rewards recall of narrow procedural specifics rather than general scientific reasoning. Relatedly, the analysis does not identify which subsets of questions humans answer reliably above chance. Another potential complication is that the GPT-4o vision-blind filter used during quality control is also one of the evaluated models in Table 2, creating a plausible circularity. If items that GPT-4o can answer without vision are preferentially removed, the remaining set could be unintentionally shaped in a way that depresses GPT-4o’s measured standing relative to other models. Finally, the quantitative distractor strategy differs from the conceptual and hypothetical cases. For quantitative questions, distractors are produced by adding Gaussian noise to the correct value (Section C), which may lead to options that are systematically less plausible than distractors crafted with a more semantics-driven approach. The paper does not provide evidence that these quantitative distractors match the difficulty level of those used in the other categories

---

> ### Author Rebuttal · Authors · 2026-03-31
>
> ## 1. JoVE videos introduce domain bias
>
> We respectfully disagree that using JoVE as the data source necessarily introduces domain bias. We argue that JoVE provides a *diverse and representative collection of real-world experimental videos*, rather than a narrow or biased distribution, for the following reasons:
>
> - **Diverse institutional sources**: JoVE videos are produced by independent research groups across universities and institutes worldwide (241 videos/papers with 881 unique affiliations), rather than a single curated production pipeline.
>
> - **Broad scientific coverage**: SciVideoBench spans four major disciplines, further divided into multiple specialized subfields (e.g., bioengineering, materials science, analytical chemistry, oncology, fluid dynamics), each involving distinct experimental paradigms and reasoning requirements.
>
> - **Heterogeneous experimental setups**: The videos exhibit substantial variation in instrumentation, procedures, and observation targets, reflecting real differences across laboratories and scientific domains.
>
> - **Natural variability in video presentation**: Since videos are independently produced, they vary in narration style, camera viewpoints, temporal structure, and laboratory environments, reducing the risk of consistent stylistic bias.
>
> Overall, JoVE serves as a high-quality and diverse source of *peer-reviewed experimental content*, enabling evaluation of visual-grounded reasoning across heterogeneous real-world scientific scenarios.
>
>
>
> ## 2. Imbalanced discipline distribution
>
> We thank the reviewer for this important observation. We report per-discipline performance in Table 2, which allows direct assessment of whether the aggregate accuracy is disproportionately shaped by the majority category (Biology).
>
> Across models, performance is not dominated by Biology despite its larger share of videos. For example, Gemini-2.5-Pro achieves comparable accuracy on Biology (64.79%), Chemistry (61.82%), and Physics (61.44%), while Medicine attains the highest accuracy (74.77%). Similar trends hold for other models.
>
> These results indicate that the aggregate metric reflects cross-disciplinary scientific video reasoning rather than being driven by the majority domain.
>
>
>
> ## 3. Low human baseline
>
> We thank the reviewer for raising this important concern. The human baseline was collected in a closed-book setting, where participants had access only to the video without external references.
>
> To ensure reasonable domain coverage, we recruited graduate students based on broad disciplinary categories (physics, biology, chemistry, medicine). However, as shown in Figure 3, each discipline in SciVideoBench contains multiple specialized subfields (e.g., fluid dynamics, materials science, molecular biology, bioengineering). As a result, even participants within a given discipline are not experts in many of the specific experimental contexts presented in the benchmark.
>
> In addition, our three question types (detailed definitions in Appendix D) require reasoning over visual evidence (e.g., interpreting mechanisms, counterfactual outcomes, or performing calculations from video-derived values), rather than recalling procedural specifics. This mitigates the risk that performance is driven by memorization instead of general scientific reasoning.
>
>
>
> ## 4. GPT-4o vision-blind filtering circularity
>
> We thank the reviewer for raising this important concern. However, we emphasize that this design does not introduce bias against GPT-4o, but rather serves as a **conservative filtering strategy** to remove questions that are trivially solvable through textual or prior knowledge alone. In other words, the filtering aims to eliminate text-only shortcuts, ensuring that the remaining questions require visual grounding.
>
>
> ## 5. Quantitative distractors via Gaussian noise
>
> We thank the reviewer for this insightful observation. While quantitative distractors are initially generated by adding Gaussian noise to the correct value, we apply an additional human verification step to ensure that all options remain physically plausible and free of artifacts such as negative masses or unrealistic magnitudes. We will clarify the distractor validation process in the final version to address this concern.

---

> > ### Author Rebuttal · Reviewer_4oDm · 2026-04-05
> >
> > I appreciate the rebuttal from the authors. But still, the scientific rigor of the low human baseline and quantitative distractor remains in question. I would like to preserve the current score.

---

> > > ### Author Response · Authors · 2026-04-05
> > >
> > > We thank the reviewer for the continued feedback and for raising this important concern.
> > >
> > > ## 1. Low human baseline
> > > We acknowledge that the current human evaluation does not include multi-pass annotations, and therefore does not explicitly quantify uncertainty at the level of individual questions. This is a limitation of the current version. At the same time, constructing a high-quality human baseline for SciVideoBench is inherently challenging. Unlike prior benchmarks that focus on a narrower domain, SciVideoBench spans multiple disciplines and fine-grained subfields (e.g., condensed matter physics vs. optics, or molecular biology vs. bioengineering). Even graduate-level participants typically do not possess expertise across all these subdomains, making it difficult to recruit human experts who can confidently answer every question. As a result, **the observed low human performance reflects not only task difficulty but also the cross-disciplinary nature of the benchmark**. The human baseline primarily serves as a difficulty indicator rather than a strict upper bound.
> > >
> > > We agree that a more rigorous human evaluation would further strengthen the benchmark. In future work, we plan to (1) recruit annotators with **more specialized domain expertise at the subfield level**, and (2) conduct **multi-pass evaluations to better estimate human consistency and uncertainty**. This will allow a more precise characterization of human performance on SciVideoBench.
> > >
> > > ## 2. Quantitative question distractor design
> > > We thank the reviewer for this important concern. We agree that **reasoning-based distractors** (e.g., derived from incorrect intermediate steps) can be more semantically aligned for quantitative questions. We will explore incorporating reasoning-based distractors in future versions to further improve alignment with semantic difficulty.

---

### Decision · Program_Chairs · 2026-04-30

**Decision:**

Accept (regular)

**Comment:**

This paper proposes SciVideoBench, a benchmark for scientific video reasoning. It received mixed reviews: Weak Reject ×2 and Accept ×2. The reviewers recognize that the dataset is rigorously constructed using a multi-stage human–AI collaborative annotation pipeline, that the three-part taxonomy of question types is well-structured, and that the dataset is challenging— even top models struggle significantly.

On the other hand, the reviewers raised concerns about potential contamination and its distinction from recent benchmarks. The authors provided a detailed rebuttal. Although some minor issues remain, most concerns have been addressed.

Considering the timely contribution of this work, I believe the strengths outweigh the weaknesses. Hence, I recommend accepting this paper.